# A Closer Look at ACE2 Signaling Pathway and Processing during COVID-19 Infection: Identifying Possible Targets

**DOI:** 10.3390/vaccines11010013

**Published:** 2022-12-21

**Authors:** Pia V. Sodhi, Francoise Sidime, David D. Tarazona, Faviola Valdivia, Kelly S. Levano

**Affiliations:** 1Ekarus, New York, NY 10023, USA; 2Science Department, Helene Fuld College of Nursing, New York, NY 10035, USA; 3Laboratorio de Infecciones Respiratorias Agudas, Centro Nacional de Salud Pública, Instituto Nacional de Salud, Lima 15072, Peru

**Keywords:** COVID-19, ACE2, RAAS

## Abstract

Since the identification of its role as the functional receptor for SARS-CoV in 2003 and for SARS-CoV-2 in 2020, ACE2 has been studied in depth to understand COVID-19 susceptibility and severity. ACE2 is a widely expressed protein, and it plays a major regulatory role in the renin–angiotensin–aldosterone System (RAAS). The key to understanding susceptibility and severity may be found in ACE2 variants. Some variants have been shown to affect binding affinity with SARS-CoV-2. In this review, we discuss the role of ACE2 in COVID-19 infection, highlighting the importance of ACE2 isoforms (soluble and membrane-bound) and explore how ACE2 variants may influence an individual’s susceptibility to SARS-CoV-2 infection and disease outcome.

## 1. Introduction

Since the first cases reported from China to the World Health Organization (WHO) between December 2019, and January 2020 [1], there has been an enormous effort to identify the mechanism of action of the agent SARS-CoV-2 (severe acute respiratory syndrome coronavirus 2). The infectivity, course, and severity of COVID-19 are affected by the emergence of new strains as well as host-associated genetic factors. The identification of these genetic factors contributing towards COVID-19 susceptibility and severity will help to identify new therapeutic targets. A key gene that constitutes a genetic risk factor for the viral infection is the ACE2 (angiotensin-converting enzyme) gene, which encodes for the entry receptor for SARS-CoV-2 into host cells [2]. In this review, we direct our focus towards ACE2, discussing its role in RAAS and during SARS-CoV-2 infection, and some reported ACE2 variants and their effects on protein stability, ligand-receptor affinity, proteolytic cleavage site, and interindividual variability in different populations. 

## 2. ACE2 Protein

### 2.1. ACE2 and RAAS

ACE2 can be described as an enzyme, a transporter, and through its role as a receptor. As an enzyme, it is an important regulator of the RAAS, a major regulator system of human physiology, controlling blood pressure, volume, and electrolytes, thus affecting the heart, vasculature, and kidney, mainly through the actions of the octapeptide hormone angiotensin II (Ang II) [3,4] (Figure 1). As expected, ACE2 is highly expressed in endothelial cells from the heart, kidney, upper airways, lungs, gut, liver, and testis [5]. Specifically, ACE2, as a zinc metallopeptidase with carboxypeptidase activity, hydrolyzes Ang II. Ang II is a vasoconstrictor that promotes inflammation and increases oxidative stress and apoptosis through the AT1 (angiotensin II type 1) receptors. ACE2, as a regulator, prevents its accumulation and thus minimizes its effect [5,6,7] (Figure 2a). When there is a reduced protein expression of ACE2 and, consequently, a buildup of Ang II, an increase in hypertension is observed, as shown in several animal models [5,8]. Additionally, ACE2 also plays an important role in glucose homeostasis. It has been shown that ACE2 overexpression in diabetic mice improves islet function and glycemic control [9,10]. 

### 2.2. ACE2 Processing during SARS-CoV-2 Infection

ACE2 acts as the receptor for SARS-CoV and SARS-CoV-2. The binding of these viruses to the membrane-bound form of the ACE2 receptor is necessary for virus internalization [11]. The reason why COVID-19 has had a much bigger global impact than SARS is due to biding affinity. In SARS-CoV-2, the region that interacts with the metallopeptidase domain of ACE2 is the receptor-binding domain (RBD) in the S protein. This binding causes structural changes in the S protein, exposing the cleavage sites at the S1/S2 or adjacent regions, which are attacked by host cellular proteases [12]. The RBD of SARS-CoV-2 has a stronger binding affinity with the ACE2 receptor due to the five out of six changes of vital amino acids, enhancing the connection with stronger hydrophobic interactions, despite the close linkage of the viruses [5]. These five variations are in the amino acids Leu455, Phe486, Gln493, Ser494, and Asn501 in SARS-CoV-2. Out of the five, positions Gln493 and Asn501 have been highlighted as the most critical amino acid residues important for van der Waals interactions and hydrogen bonding [13].

In addition to ACE2, SARS-CoV-2 requires further processing to enter the host cell. Transmembrane Serine Protease 2 (TMPRSS2), a serine endopeptidase primes the S protein. This entails the cleaving of the S protein at the subunit 1 and 2 sites, as well as at the S2 site [14]. This allows for a fusing of both the cellular and viral membranes. SARS-CoV-2 also uses another protease for S protein priming, the endosomal cysteine proteases cathepsin B and L (CatB/L). SARS-CoV requires TMPRSS2 processing for viral spread, while CATB/L activity can be dispensable. This is not the case for SARS-CoV-2, which requires both TMPRSS2 and CATB/L activity for viral entry [14]. There are also data that suggest ACE2 is shed from membranes with the help of TMPRSS2, which leads to membrane fusion and the cellular uptake of the virus [15,16] (Figure 2b). 

### 2.3. ACE Isoforms

ACE2 is a type I transmembrane protein of 805 amino acids containing an ectodomain (enzyme) and a C-terminal transmembrane anchor [17]. The extracellular N-terminal domain contains a zinc metallopeptidase catalytic site and the spike protein binding site where SARS-CoV and SARS-CoV-2 bind (amino acids 1–740), a short transmembrane domain (amino acids 741–763), and a C-terminal domain facing the cytosol [18,19,20] (Figure 3). ACE2 has two functional forms, as a membrane bound receptor and as a soluble form (sACE2) with 555 amino acids [21,22]. The enzyme ADAM-17 (a disintegrin and metallopeptidase domain 17) or TACE (tumor necrosis factor-converting enzyme) is responsible for cleaving ACE2 at amino acids 716–741 [23,24]. ADAM-17, a type I transmembrane protein belonging to the family of zinc-dependent metalloproteases, catalyzes ACE2 ectodomain shedding, compromising the role of ACE2 in regulating the RAAS. sACE2 maintains carboxypeptidase activity and thus the ability to bind to the RBD region in the viral S protein. In a study by Haga et al., in 2008, both the cytoplasmic tails of ADAM-17 and of ACE2 were shown to be necessary for SARS-CoV to infect host cells; however, the underlying mechanism is still unknown [25,26]. During SARS-CoV-2 infection, the binding of the ACE2 receptor to the viral S1 protein promotes the cleavage of the ACE2 ectodomain by ADAM-17 and the intracellular C-terminal domain by TMPRSS2, thus facilitating SARS-CoV-2 entry [27] (Figure 2c). ACE2 is internalized with the viral particles into endosomes. Together, ACE2 processing and internalization reduce its expression in the membrane, affecting the regulation of Ang II and promoting RAAS imbalance and the activation of the AT1 receptors [7,28]. In addition, ADAM-17 expression is also increased by ACE2 and SARS-CoV-2 internalization, further increasing ACE2 ectodomain proteolytic cleavage and sACE2 production.

sACE2 can be beneficial for preventing COVID-19. Many studies have reported its preventive role as a therapeutic agent due to its ability to bind to circulating SARS-CoV-2 and thus block viral entry [29,30,31] (Figure 2c). However, other researchers view sACE2 as another key to enter and infect non-ACE2-expressing cells. Yeung et al. showed that sACE2 facilitates SARS-CoV-2 entry through receptor-mediated endocytosis and thus enables its entrance in various tissues [32]. 

## 3. ACE2 Variants and COVID-19 Susceptibility

It is crucial to identify at risk individuals in specific populations. The various ACE2 variants differ in how they affect a given population’s susceptibility to SARS-CoV-2. Globally, in 2021 the WHO has reported a total of 178,837,204 million cases with fewer cases reported in Africa and Western Pacific [33]. ACE2 genetic variation, especially deleterious missense variants in ACE2 flexible regions (regions that change between an open and close state when bound to the virus), may affect its function and structure, and thus may alter its affinity towards SARS-CoV-2 [34]. The ACE2 gene is a highly polymorphic gene [28,35] containing 18 exons and 20 introns and is located in the chromosome Xp22 (different location from the ACE gene, which encodes for the ACE protein) [36]. ACE2 variants that are only present in specific populations are presented in Table 1. There are some variants, such as rs1244687367 (I21T), that have been shown to improve binding and hence susceptibility to the virus in all populations and regions [37]. From previously reported structural data, different research groups have predicted the effect of various ACE2 variants on ACE2–SARS-CoV-2 interaction and thus host susceptibility. Some of these predictions were further confirmed using biochemical assays [38].

Some variants differ even within populations. In African and African American populations, the variant rs73635825 (S19P) has been shown to both enhance affinity for the S protein of the SARS-CoV-2 and in some provide a lower binding affinity for the spike protein due to levels of resistance. This variant is located at a crucial site where the virus S-protein interacts, at the beginning of the helix Ser19-Ile54, helping stabilize the helical structure through hydrogen bonding and hydrophilic interactions. Thus, the change from Serine to Proline (having poor helix-forming properties) could lead to either breaks or kins in the helix structure [38].

In American populations, there are two predominant variants, rs781255386 (T27A) and rs924799658 (F40L) that have been found to increase binding affinity and thus increasing susceptibility. With variant T27A, the change from Threonine to Alanine leads to an increased hydrophobic environment that could explain an increase in binding affinity due to this mutation [38].

In European non-Finnish populations, two variants, rs778030746 (I21V) and rs756231991 (D23K) have been associated with enhance binding and increase susceptibility. In contrast, two other variants in these same populations, rs1192192618 (Y50F) and rs1325542104 (M62V), have exhibited lower binding affinity to the SARS-CoV-2 spike protein. In South Asian populations, the variant rs760159085 (N51D) has been shown to have a lower affinity for the coronavirus. While many variants have been shown to affect specific populations, there are many whose effects are not yet known. 

Although limited, ACE2 variants among different populations could partially begin to explain differences in COVID-19 susceptibility. 

## 4. Conclusions

In this review, we highlight the role of ACE2 as a RAAS regulator and SARS-CoV-2 receptor (Figure 2). We describe the role of the two forms of ACE2, mACE2 and sACE2. The interconnection between SARS-CoV-2, mACE2, sACE2, and ANG II should be further studied in relation to COVID-19 severity. Additionally, we discuss ACE2 variants that have been identified to be specific to different populations, thus shedding light onto interindividual variability in different populations. 

Various studies have suggested ACE2 as a possible biomarker and therapeutic target for fighting COVID-19. Especially due to its role as a RAAS regulator, sACE2 could act as a biomarker for hypertension, inflammatory diseases, and heart failure [20,42]. It could also act as a biomarker for COVID-19 susceptibility and progression [43], especially in specific populations. As a therapeutic target, the identification of ACE2 variants and their roles are crucial. Some of the strategies that are currently being used and proposed include the blockage of the ACE2–SARS-CoV-2 binding using specific ligands or antibodies and sACE2 recombinant proteins to neutralize SARS-CoV-2. 

## Figures and Tables

**Figure 1 vaccines-11-00013-f001:**
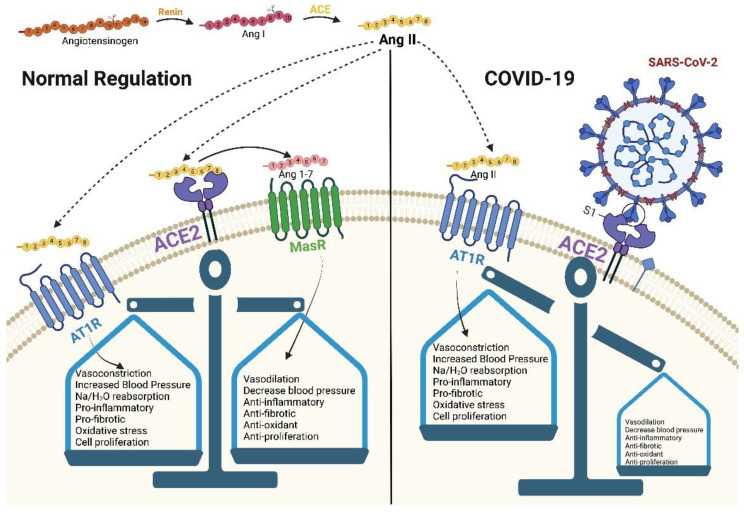
SARS-CoV-2 blocks ACE2 causing RAAS dysregulation. Created with BioRender.com.

**Figure 2 vaccines-11-00013-f002:**
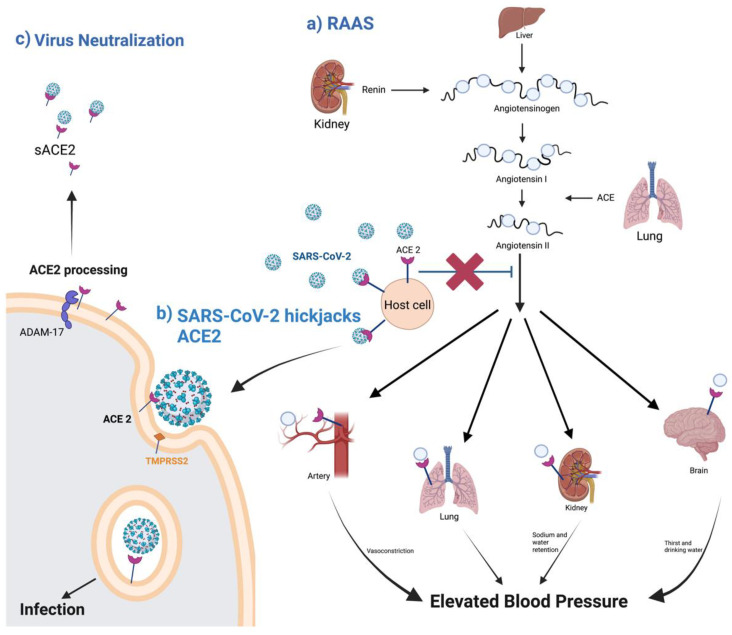
**A schematic diagram of ACE2 pathways.** (**a**) the regulation of the renin–angiotensin system. (**b**) ACE2 as SARS-CoV-2 receptor, leading to the internalization and infection of the virus. (**c**) ACE2 processing by ADAM-17, leading to the release of sACE2 and the possible blockage of SARS-CoV-2 infection. Created with BioRender.com.

**Figure 3 vaccines-11-00013-f003:**
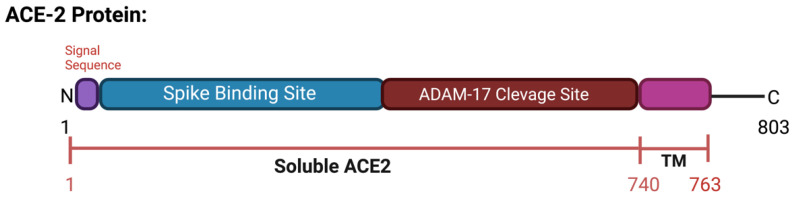
Schematic representation of the ACE2 protein. Includes spike binding site and ADAM-17 cleavage site and soluble ACE2. Created with BioRender.com.

**Table 1 vaccines-11-00013-t001:** ACE2 variants: interindividual variability in different populations.

	dbSNP ID	Substitution	Description	References
AFR	rs73635825	S19P	Enhances affinity for spike protein second most common ACE2 variants Binds SARS-CoV-2 spike more strongly than reference ACE2. Lowers binding affinity for SARS-CoV-2 spike protein likely to provide some level of resistance against SARS-CoV-2 attachment to ACE2	[39]
				[40]
AMR	rs781255386	T27A	Enhances binding affinity to SARS-CoV-2 S protein Found in the binding region	[37,41]
	rs924799658	F40L	Increases spike binding Relatively rare	[39]
NFE	rs778030746	I21V	Found in the binding region	[37]
	rs756231991	D23K	Found in the binding region	[37]
ALL Al	rs1244687367	I21T	Improves binding	[37]

## Data Availability

Not applicable.

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
