# Peer review of "A Closer Look at ACE2 Signaling Pathway and Processing during COVID-19 Infection: Identifying Possible Targets"

_vaccines, 2022, doi:10.3390/vaccines11010013_

Round 1
Reviewer 1 Report
This manuscript describes a review of the various ways by which ACE2 is processed and the role of different variants in COVID-19 disease susceptibility. The information given is exhaustive and is a valuable addition to the field. I have some minor suggestions that I have given below.
1) Since this review is about ACE2, more information is needed about the receptor especially regarding full details of its structure and binding domain. A schematic figure to illustrate this would be useful.
2) What are the roles of variants in ACE2 functions. Do they enhance or reduce their functions> The authors should give more details regarding thisin their manuscript
3) A schematic figure of the various pathways used by ACE2 for its functions and how SARS COV-2 hijacks that pathway needs to be shown.
4) Details of how the different variants change the affinity for SARS-COV-2 is not clear. More information should be given in this regard.
Author Response
We want to begin by thanking the reviewers for their suggestions to improve our article. We have made the corresponding changes in accordance to their suggestions as follows:
Point 1: Since this review is about ACE2, more information is needed about the receptor especially regarding full details of its structure and binding domain. A schematic figure to illustrate this would be useful.
Response 1: We have added to following: "The extracellular N-terminal domain contains a zinc metallopeptidase catalytic site and the spike protein binding site where SARS-CoV and SARS-CoV-2 bind, a short transmembrane domain and a 44 amino acid C-terminal domain facing the cytosol [17, 18] (Figure 2).
Additionally we have complemented as suggested with a figure illustrating the structure and domains of ACE2.
Point 2: What are the roles of variants in ACE2 functions. Do they enhance or reduce their functions? The authors should give more details regarding this in their manuscript.
Response 2: To address this point, we had stated the following in the article: " ACE2 genetic variation, especially deleterious missense variants in ACE2 flexible regions (regions that change between an open and close state when bound to the virus), may affect its function and structure, and thus may alter its affinity towards SARS-CoV-2 [33]. "
To further complement and add more details currently available, we have added the following: " This variant is located at a crucial site where the virus S-protein interacts, at the beginning of the helix Ser19-Ile54, helping stabilize the helical structure through hydrogen bonding and hydrophilic interactions. Thus, the change from Serine to Proline (having poor helix-forming properties) could lead to either breaks or kins in the helix structure [36]." and "With variant T27A, the change from Threonine to Alanine leads to an increased hydrophobic environment that could explain an increase in binding affinity due to this mutation [37]."
For the other variants, there is no additional information.
Point 3: A schematic figure of the various pathways used by ACE2 for its functions and how SARS COV-2 hijacks that pathway needs to be shown.
Response 3: As suggested, we have added a figure (Figure 3) highlighting ACE2 functions and pathways.
Point 4: Details of how the different variants change the affinity for SARS-COV-2 is not clear. More information should be given in this regard.
Response 4: To address this point and provide more information, we have added the following: " From previously reported structural data, different research groups have predicted the effect of various ACE2 variants on ACE2-SARS-CoV-2 interaction and thus host susceptibility. Some of these predictions were further confirmed using biochemical assays."
Reviewer 2 Report
Although it thinks that there is a review that has been written well and summarizes useful information about SARS-CoV-2 and ACE2, but I believe that some explanations are missing and the issue is suddenly cut.
For example, sufficient details have not been given about ACE2's role as a possible biomarker and therapeutic target. Which therapeutics were used for this infection? What goal are these for? What are their activities?
Additionaly, some minor revisions are listed below.
Although it thinks that there is a review that has been written well and summarizes useful information about SARS-CoV-2 and ACE2, but I believe that some explanations are missing and the issue is suddenly cut.
For example, sufficient details have not been given about ACE2's role as a possible biomarker and therapeutic target. Which therapeutics were used for this infection? What goal are these for? What are their activities?
Some minor revisions:
Line 35- ….its latest role-……it should be corrected as “role”.
Figure 1- Who was taken from? Citation? Was it drawn by the authors? Which program did the authors used this figure? It should be explained.
Line 42- In its place AT abbreviation should be written AT1.
Lines 59-60- due to the 5 out of 6 changes of vital amino acids--- Which amino acids? And their importance? It should be clarified.
Line 63- TMPRSS2 (Transmembrane Serine Protease 2) should be corrected as “Transmembrane Serine Protease 2 (TMPRSS2)”.
Line 85- in its place TMPRSS2 (transmembrane protease serine protease 2) should be written only TMPRSS2.
Line 65- its place S2’ should be “S2”.
Line 98- Subtitle- 3. ACE2 variants and COVID-19 susceptibility
In this section, only 3 variations are explained. It was found to be inadequate to establish this connection.
Line 99- in its place at-risk should be written “at risk”.
Line 101- a total of 178,837,204 cases,,,, in which date?
Line 121- “for the Spike Protein of the coronavirus” should be corrected “S protein of the SARS-CoV-2”.
Lines between 107 and 121- Which literature was taken from these sentences? Citations?
Author Response
We want to begin by thanking the reviewer for their suggestions to help us improve our article.
Point 1: Although it thinks that there is a review that has been written well and summarizes useful information about SARS-CoV-2 and ACE2, but I believe that some explanations are missing and the issue is suddenly cut. For example, sufficient details have not been given about ACE2's role as a possible biomarker and therapeutic target. Which therapeutics were used for this infection? What goal are these for? What are their activities?
Response 1: To address the following, we have added the following:
" Various studies have suggested ACE2 as a possible biomarker and therapeutic target for fighting COVID-19. Especially due to its role as a RAAS regulator, sACE2 could act as a biomarker for hypertension, inflammatory diseases, and heart failure [20, 43]. It can also act as a biomarker for COVID-19 susceptibility and progression [44] especially in specific populations. As a therapeutic target, the identification of ACE2 variants and their roles are crucial. Some of the strategies that are currently being used and proposed include the blockage of ACE2-SARS-CoV-2 binding using specific ligands or antibodies and sACE2 recombinant proteins to neutralize SARS-CoV-2."
Point 2: Additionaly, some minor revisions are listed below. Some minor revisions:
Line 35- ….its latest role-……it should be corrected as “role”.
Response 2: It has been changed as suggested.
Point 3: Figure 1- Who was taken from? Citation? Was it drawn by the authors? Which program did the authors used this figure? It should be explained.
Response 3: This figure was made by the authors using Biorender. We have added the following to the figure: " Created with BioRender.com."
Point 4: Line 42- In its place AT abbreviation should be written AT1.
Response 4: It has been changed as suggested.
Point 5: Lines 59-60- due to the 5 out of 6 changes of vital amino acids--- Which amino acids? And their importance? It should be clarified.
Response 5: To address the suggestion, we have added the following: "These five variations are in the amino acids Leu455, Phe486, Gln493, Ser494 and Asn501 in SARS-CoV-2. Out of the five, positions Gln493 and Asn501 have been highlighted as the most critical amino acid residues important for van ser Waals interactions and hydrogen bonding [13]."
Point 6: Line 63- TMPRSS2 (Transmembrane Serine Protease 2) should be corrected as “Transmembrane Serine Protease 2 (TMPRSS2)”.
Response 6: It has been changed as suggested.
Point 7: Line 85- in its place TMPRSS2 (transmembrane protease serine protease 2) should be written only TMPRSS2.
Response 7: It has been changed as suggested.
Point 8: Line 65- its place S2’ should be “S2”.
Response 8: It has been changed as suggested.
Point 9: Line 98- Subtitle- 3. ACE2 variants and COVID-19 susceptibility
In this section, only 3 variations are explained. It was found to be inadequate to establish this connection.
Response 9: To address this, we have added more details.
"From previously reported structural data, different research groups have predicted the effect of various ACE2 variants on ACE2-SARS-CoV-2 interaction and thus host susceptibility. Some of these predictions were further confirmed using biochemical assays [38].
Some variants can differ even within populations. In African and African American populations, the variant rs73635825 (S19P) has been shown to both enhance affinity for the S protein of the SARS-CoV-2 and in some provide a lower binding affinity for the spike protein due to levels of resistance. This variant is located at a crucial site where the virus S-protein interacts, at the beginning of the helix Ser19-Ile54, helping stabilize the helical structure through hydrogen bonding and hydrophilic interactions. Thus, the change from Serine to Proline (having poor helix-forming properties) could lead to either breaks or kins in the helix structure [38].
In American populations, there are two predominant variants, rs781255386 (T27A) and rs924799658 (F40L) that have been found to increase binding affinity and thus increasing susceptibility. With variant T27A, the change from Threonine to Alanine leads to an increased hydrophobic environment that could explain an increase in binding affinity due to this mutation [39].
In European Non-Finnish populations, two variants, rs778030746 (I21V) and rs756231991 (D23K) have been associated with enhance binding and increase susceptibility. In contrast, two other variants in these same populations, rs1192192618 (Y50F) and rs1325542104 (M62V), have exhibited lower binding affinity to SARS-CoV-2 Spike protein. In South Asian populations, the variant rs760159085 (N51D) has been shown to have a lower affinity for the coronavirus. While many variants have been shown to affect specific populations, there are many whose effects are not yet known.
Although limited, ACE2 variants among different populations, could partially begin to explain differences in COVID-19 susceptibility."
Point 10: Line 99- in its place at-risk should be written “at risk”.
Response 10: It has been changed as suggested.
Point 11: Line 101- a total of 178,837,204 cases,,,, in which date?
Response 11: To address this, the year was added in the text.
"The various ACE2 variants differ in how they affect a given population’s susceptibility to SARS-CoV-2. Globally, in 2021 the WHO has reported a total of 178,837,204 million cases with fewer cases reported in Africa and Western Pacific [33]. "
Point 12: Line 121- “for the Spike Protein of the coronavirus” should be corrected “S protein of the SARS-CoV-2”.
Response 12: It has been changed as suggested.
Point 13: Lines between 107 and 121- Which literature was taken from these sentences? Citations?
Response 13: As suggested the corresponding references has been added.
Round 2
Reviewer 1 Report
I agree with the changes made. The manuscript is ready for publication
Reviewer 2 Report
I found that the authors corrected the deficiencies identified by me in the article and answered my critiques. It is acceptable for this reason.